# Predicting from Strings: Language Model Embeddings for Bayesian Optimization

## Abstract

Bayesian Optimization is ubiquitous in the field of experimental design and black-box optimization for improving search efficiency, but has been traditionally restricted to regression models which are only applicable to fixed search spaces and tabular input features. We propose *Embed-then-Regress*, a paradigm for applying in-context regression over string inputs, through the use of string embedding capabilities of pretrained language models. By expressing all inputs as strings, we able to perform general-purpose regression for Bayesian Optimization over different search domains such as traditional and combinatorial optimization, obtaining comparable results to state-of-the-art Gaussian Process-based algorithms.

## 1 Introduction

A fundamental component of all *value-based* search methods is regression, in which proposed solutions are filtered by predictions on their performance, before evaluation. By utilizing an accurate regression model, or *regressor*, along with balanced explore-exploit mechanisms, large improvements to the sample complexity of search have been widely possible. However, many regression methods so-far have been task-specific, due to the reliance of modelling assumptions and dimensionality constraints. Learning-based regression methods are particularly susceptible to this issue, due to their reliance on fixed-length tensors for input representation.

Recent progress in large language models (LLMs) have demonstrated the flexibility and versatility of representation of information as strings, which allow for a wider range of data formats to be encoded for subsequent processing. The potential of LLMs for universal learning-based regression is considerable, allowing for regressors that can be generalized across multiple tasks, thereby mitigating the task-specific limitations of current methods.

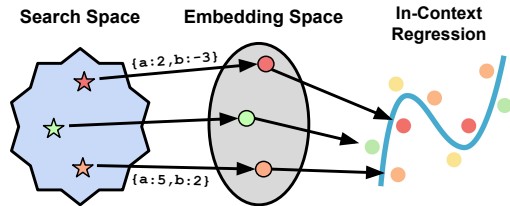

Figure 1: Using language models, we embed string representations of search space candidates as features for downstream regression.

In this work we focus on improving the flexibility of regressor-guided search methods through the use of LLM-based embeddings, which map arbitrary strings to fixed-length vectors to be used in downstream tensor-based regressor models, such as an in-context learning (ICL) based Transformer. Specifically, our contributions are:

- We describe our framework, "embed-then-regress" which uses a language model to embed a string representation of a trial to be used as a single token feature for a in-context regressor, such as a Transformer-based neural process.

- By pretraining this regressor at scale over a large variety of offline evaluation data, we can achieve uncertainty-aware numeric predictions over objective functions from unseen tasks.

- After augmenting the framework with explore-exploit techniques, we achieve competitive optimization results over a variety of optimization tasks, including traditional and combinatorial optimization.

## 2 RELATED WORK AND MOTIVATION

Bayesian Optimization refers to a class of techniques which use regressors for solving blackbox optimization problems, by suggesting best candidates according to an explore-exploit tradeoff. For a given objective, the speed of optimization relies heavily on the regressor's underlying *prior*, or assumptions about the nature of the objective, such as its smoothness and landscape. Largely dominated by the use of Gaussian Process (GP) regressors, the field of Bayesian Optimization has thus seen a rise in works (Wang et al., 2024; Fan et al., 2024) which seek to learn better prior GP hyperparameters such as length-scales and kernel amplitudes based on offline pretraining or manually designed feature representations for combinatorial objects (Deshwal et al., 2023), while keeping underlying kernel definitions fixed.

Numerous end-to-end neural network-based approaches such as the use of attention mechanisms and Transformers (Vaswani et al., 2017) have been introduced to allow more learnable behaviors, and we refer the reader to (Song et al., 2024b) which provides a general reference on their use for blackbox optimization. Relevant to our particular case of regression, works such as (Nguyen & Grover, 2022; Garg et al., 2022) demonstrated the benefits of using raw Transformers as in-context regression models, or *neural processes*, with others (Bai et al., 2023; Zhang et al., 2024) establishing provable guarantees. Similarly, (Müller et al., 2022) demonstrated that Transformers trained on synthetic data as "prior-fitted networks" are capable of Bayesian inference, leading to their use in Bayesian optimization (Müller et al., 2023; Nguyen & Grover, 2024).

Unfortunately, as both raw Transformers and GPs require fixed dimensional features, this limits their applications to inputs expressable as tabular features for e.g. hyperparameter tuning, or task-specific embeddings for e.g. chemistry (Maus et al., 2022). Further works have attempted to improve the flexibility of regression-modeling through the use of token-based representations, which allows regressors to be simultaneously used over various types of input formats. Since the context-window of Transformers still remain the most expensive limitation, a useful organization of recent works can be based on their treatment of the total sequence length, roughly equal to:

$$\text{(number of trials)} \times \text{(average trial token length)} \tag{1}$$

Among sequence-to-sequence methods which pretrain for blackbox optimization specifically, (Chen et al., 2022) uses custom tokenizations to minimize trial token length in order to maximize trial count. However, this is restricted to very constrained search spaces (e.g. flat hyperparameter spaces), and lacks flexibility in utilizing arbitrary forms of data. In contrast, (Song et al., 2024a) allows arbitrary string representations, but the large token length of each trial severely limits the trial count allowed in the context window, forcing the use of alternative but tedious methods of absorbing online data, such as inference-time fine-tuning.

Other methods (Vacareanu et al., 2024) use text-to-text chat-based services such as ChatGPT (OpenAI, 2022) and Gemini (Google, 2024) to demonstrate their emergent capabilities for in-context regression, but such methods lack the ability to pretrain over large amounts of offline evaluations. Efforts in ICL-based reward modeling with chat-based LLMs (Lightman et al., 2024) allow fine-tuning over chains of thought, but have only used coarse discretized scores of e.g. $\{-1, 0, 1\}$ rather than highly precise numeric predictions over vastly different scales.

For optimization, we require a regressor with all of the following capabilities:

- Pretrainable over offline evaluations to allow effective meta-learning.
- Flexible representation of inputs with raw strings for application in multiple domains.
- Allow long-range in-context regression using multiple previous evaluations.
- Production of precise numeric predictions over diverse objective scales.

This naturally leads to the use of embedding-based methods which can compress any string representation of a trial into a feature vector, using only a single unit of sequence length when sent to a ICL model such as a raw Transformer. This can be seen as a form of encoding and decoding, which has been shown to produce competitive results over pure language modeling tasks (Raffel et al., 2020).

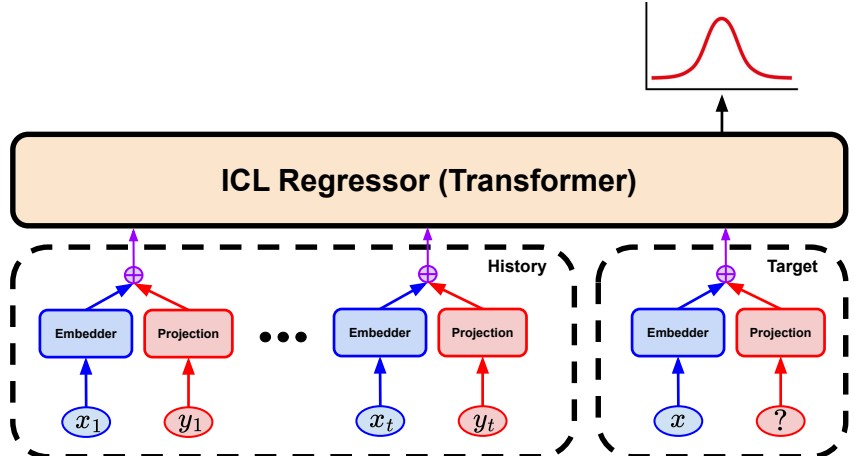

Figure 2: Overview of our model.

## 3 METHOD

### 3.1 PRELIMINARIES

Let $f : \mathcal{X} \to \mathbb{R}$ be a real-valued function over a search space $\mathcal{X}$. The goal of blackbox optimization is to produce an $x^*$ which maximizes $f$:

$$x^* = \arg\max_{x \in \mathcal{X}} f(x) \tag{2}$$

We define a *regressor* as a prediction model which can output a distribution of prediction values for $f(\cdot)$ over a query point $x$, given the history $\{x_s, y_s\}_{s=1}^t$ of trajectory of $t$ evaluations over $f$ so far. Such regressors may also be *learnable* over additional offline data besides the given history.

During inference, the regressor may be turned into an acquisition function $a : \mathcal{X} \to \mathbb{R}$ to represent explore-exploit tradeoffs. We assume the existence of a (potentially task-dependent) *acquisition optimizer* which can quickly and cheaply sample suggestions $x \in \mathcal{X}$, usually in the form of an evolutionary algorithm. The history-dependent acquisition $a_{t+1}(\cdot)$ may thus be used to filter out poor samples, or used in an entire Bayesian optimization loop in which the acquisition optimizer is used to find $x_{t+1} := \arg\max_{x \in \mathcal{X}} a_{t+1}(x)$ as the next $x$-proposal.

### 3.2 IN-CONTEXT TRANSFORMER REGRESSOR

An embedding-based regressor uses an *embedder* $\phi : \mathcal{X} \to \mathbb{R}^d$ to map a suggestion $x$ to a fixed-length representation $\overline{x} \in \mathbb{R}^d$, which can then be used as a regular feature vector for a numeric regression model. A *string-based* embedder first represents $x$ as a string, which is then passed to a language model for embedding. We specifically use the typical definition of language model embedding, in which we apply a forward pass of the underlying model (encoder or decoder) on the (tokenized) string representation to obtain all token logits in $\mathbb{R}^{L \times d}$, and then pool across the length axis to obtain a vector in $\mathbb{R}^d$. We discuss specific string representations in our experiments in Section 4.

For our underlying regression model, we then use an additional Transformer (Vaswani et al., 2017), by sending in as input sequence $(\overline{x}_1 \oplus \overline{y}_1), \ldots, (\overline{x}_t \oplus \overline{y}_t)$ where $\overline{y} \in \mathbb{R}^d$ is the feature representation of the float $y$ after applying a trainable projection, and $\overline{x} \oplus \overline{y}$ is the trial representation expressed as the concatenation of $\overline{x}$ and $\overline{y}$.

In order to obtain a prediction for a query point $x$, we may then further append a query $(\overline{x} \oplus \overline{0})$ to the history input sequence where $\overline{0}$ is a dummy value, and following a forward pass of the Transformer where $(\overline{x} \oplus \overline{0})$ attends to all previous trials, post-process the corresponding $t+1$-th output feature with a parametric output distribution over $\mathbb{R}$. For our case we assume a Gaussian $\mathcal{N}(\mu_{t+1}(x), \sigma_{t+1}^2(x))$

using mean and standard deviation output heads. The key components to our method can be summarized in Figure 2.

Additional techniques below are utilized to stabilize training and prediction:

**Parallel Predictions:** In order to simultaneously predict over a given set of $k$ *target* points $x_{t+1}, \ldots, x_{t+k}$, we additionally append $(\overline{x}_{t+1} \oplus \overline{0}), \ldots, (\overline{x}_{t+k}, \overline{0})$ to the history sequence and generate a custom attention pattern of shape $(t+k) \times t$ where all tokens to attend to the history while no tokens attend to the targets. This allows an efficient parallel modeling of $p(y_{t+i} \mid x_{t+i}, \{x_s, y_s\}_{s=0}^t)$ for $1 \leq i \leq k$ which speeds up training when computing a summed loss over multiple target predictions.

**$y$-Normalization:** Depending on the function, $y$-value outputs may contain a wide variety of scales, and need to be normalized properly. We can define our normalization procedure parameterized by a history of objectives $(y_1, \ldots, y_t)$, also applicable to incoming target values. These steps consist of, in order: (1) Shifting objectives to have zero mean and divide by standard deviation. (2) Reducing harmful effects of bad outliers by fitting the "bad half" of objectives $\{y_i \leq y_{\mathrm{median}}\}$ to a normal curve, using percentiles as z-scores. (3) Linearly scale $y \leftarrow \frac{y - y_{\min}}{y_{\max} - y_{\min}}$ which ensures all historical $y$-values within $[0, 1]$ and apply additional damping (e.g. sigmoid or log transform) to target values significantly outside this range.

**Encoding Metadata:** Many times there may be a *metadata $m$* associated to an objective $f$, which provides useful prediction information on the behavior of $f$ or simply can inform the model of a new objective or search space. We can also embed this metadata as an additional feature $\overline{m}$, which is concatenated to every $\overline{x}$ similarly to standard encoder-decoder techniques (Raffel et al., 2020).

### 3.3 PRETRAINING AND INFERENCE

Denote a task $\mathcal{T} = (f, \mathcal{X})$ as a specific objective function over a particular search space.

**Pretraining:** We assume a collection of offline *training tasks* $\{\mathcal{T}_1, \mathcal{T}_2, \ldots\}$, with different search spaces and objective functions, with each task containing its own collection of evaluated trials $\{x_s, y_s\}_{s=1}^T$ where $T$ is the (potentially task-specific) offline trajectory length.

While the embedder is frozen, we pretrain the weights $\theta$ of the ICL regression Transformer, over all such offline evaluation data. Each training example consists of a sampled task and history cutoff length $t' \in [0, T)$ so that $\{x_s, y_s\}_{s \leq t'}$ is considered a history, while $\{x_{t'+i}, y_{t'+i}\}_{t'+i}^T$ are target points, with the loss computed as the sum of prediction losses over all targets, i.e.

$$\sum_{i=1}^{T-t'} \ell_\theta(x_{t'+i}, y_{t'+i}; \{x_s, y_s\}_{s=1}^{t'}) \tag{3}$$

where $\ell_\theta(x, y; \{x_s, y_s\}_{s=1}^t)$ is the negative log-likelihood using our Gaussian output distribution, of predicting $y$ given $x$ and history $\{x_s, y_s\}_{s=1}^t$.

**Inference:** At inference, we use our mean and deviation head to form a UCB-based acquisition $a_{t+1}(x) = \mu_{t+1}(x) + \sqrt{\beta} \cdot \sigma_{t+1}(x)$ where $\sqrt{\beta}$ is a problem-dependent constant. We use a (potentially domain-dependent) zeroth-order optimizer such as evolutionary search to maximize this acquisition, and thus only require forward passess, although gradient-based acquisition maximization is possible with soft-prompt optimization techniques (Lester et al., 2021).

Since there may be distributional shifts for parameter names encountered between pretraining and inference, we may either apply data augmentation by randomizing parameter names during pretraining, or transform the search space during inference to match those encountered in pretraining.

### 3.4 MODEL DETAILS

In this paper, to demonstrate the validity of our approach on relatively low compute budgets, we intentionally use relatively smaller language model embedder sizes in comparison to the larger GPT (OpenAI, 2023) or Gemini (Google, 2024) family of models. Specifically, we use a pretrained T5-XL encoder (1B parameters), based on the encoder-decoder T5-family of models (Raffel et al., 2020). Along with only 8 layers of the underlying regression Transformer, this leads to a maximum

required training budget of approximately 16 GPUs for training and 1 GPU for inference, possible with most academic budgets.

The cheap inference cost is also necessary when the acquisition function may be called thousands of times by a zeroth-order acquisition optimizer per candidate proposal. It is worth noting that time and memory complexity costs may even further be reduced using efficient Transformers (Tay et al., 2022). Faster embedders lead to large constant factor reductions, while faster regressors can lead to linear $\widetilde{O}(t)$ complexities with respect to the number of trials.

Appendix A contains all details with respect to model sizes, training details, and hyperparameters used.

## 4 EXPERIMENTS

### 4.1 END-TO-END BLACKBOX OPTIMIZATION

To emphasize the broad applicability of our *Embed-then-Regress* method, we do not focus on achieving state-of-the-art results compared to domain-specific baselines, but rather demonstrate its effectiveness across a variety of tasks. Improvements within specific domains are left for future work. We evaluate the performance of our algorithm on various problems consisting of traditional and combinatorial objectives, with their exact details in Appendix B.

**Traditional Optimization:** In common optimization scenarios, the search space is a flat Cartesian product of float and categorical parameter types. Our string-based regression will represent each $x$ with standard JSON over the dictionary mapping parameter names to values, e.g. for a search space with two parameters, one continuous named `p0` and another integer `p1`, the string representation for an example trial would be $\{$`"p0":0.3,"p1":4`$\}$.

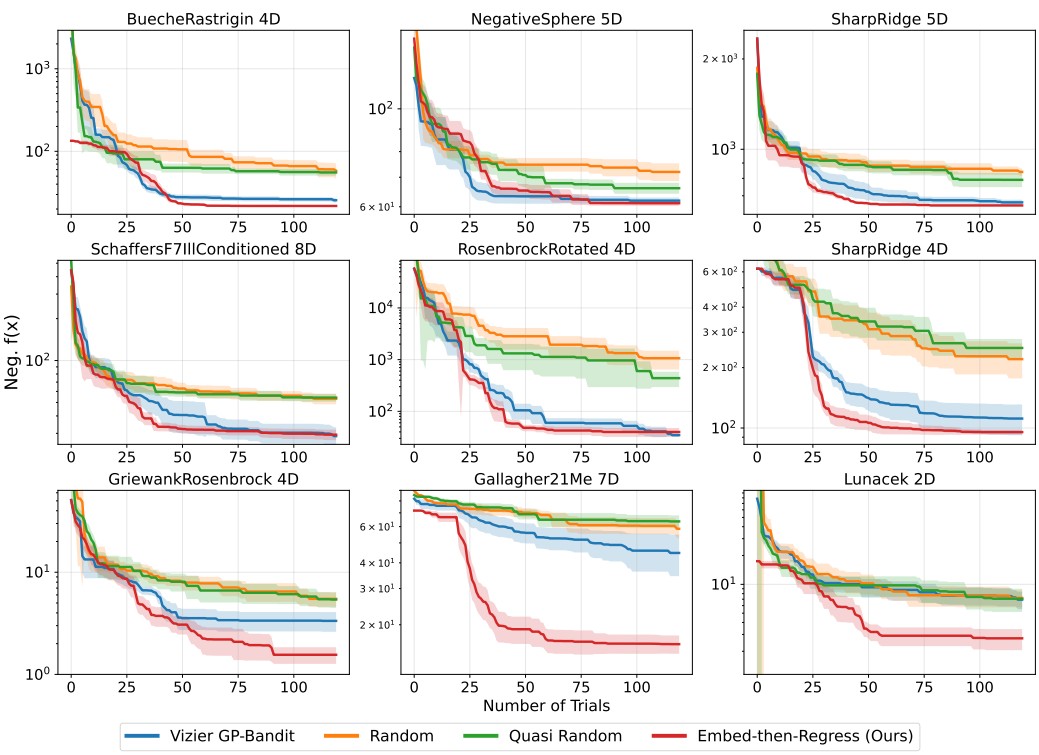

Figure 3: (↓) Lower is better. Median optimality gap curves across 9 randomized test functions, some with non-continuous parameters. Note: y-axis is log-scaled to depict clearer separation between baselines.

We benchmark over the Blackbox Optimization Benchmarking (BBOB) suite (ElHara et al., 2019), one of the most widely used synthetic function benchmarks, containing 24 different objectives over continuous search spaces. In order to have a notion of offline "training" data and unseen "test" functions to be optimized, we split the original functions across each landscape type (Separable, Ill-Conditioned, etc.) into training and test sets, and additionally apply randomized transformations (e.g. shifting, rotating, discretizing, increasing/decreasing dimensions) over all objectives to induce non-continuous search spaces with categorical parameters and avoid overfitting. Evaluations were uniformly sampled over the search space.

As a traditional GP baseline, we use the industry-grade UCB-based Bayesian Optimization method "GP-Bandit" (Song et al., 2024c) from Open Source Vizier (Song et al., 2022). In order to control for confounding factors affecting performance, our method uses the same "Firefly" acquisition optimizer as Vizier's, with the same evaluation budget. In Figure 3, we find that Embed-then-Regress is generally comparable with and interestingly can even significantly outperform GP-Bandit in a few cases.

**Combinatorial Optimization:** We further benchmark over combinatorial objectives whose search spaces are typically difficult to regress over. Many of these can be found in common operations research literature, e.g. permutation-based (Travelling Salesman, Quadratic Assignment, Flowshop Scheduling, and N-Queens), and choice-based (submodular maximization problems such as covering and log-determinant functions).

Each of these problems can be parameterized by a set of coefficients (e.g. city locations for Travelling Salesman, matrix entries for log-determinant). Note that we are in the *bandit* setting, in which these coefficients are hidden from the algorithm and the only feedback is the final objective. Similar to before, we thus can also generate offline pretraining data by randomizing these coefficients and problem sizes, and evaluating over random candidates. For our string regression, we may simply use JSON over indices; e.g. `[2,0,3,1]` for a permutation space of size 4, e.g. `[1,3]` for a $\binom{4}{2}$ choice space.

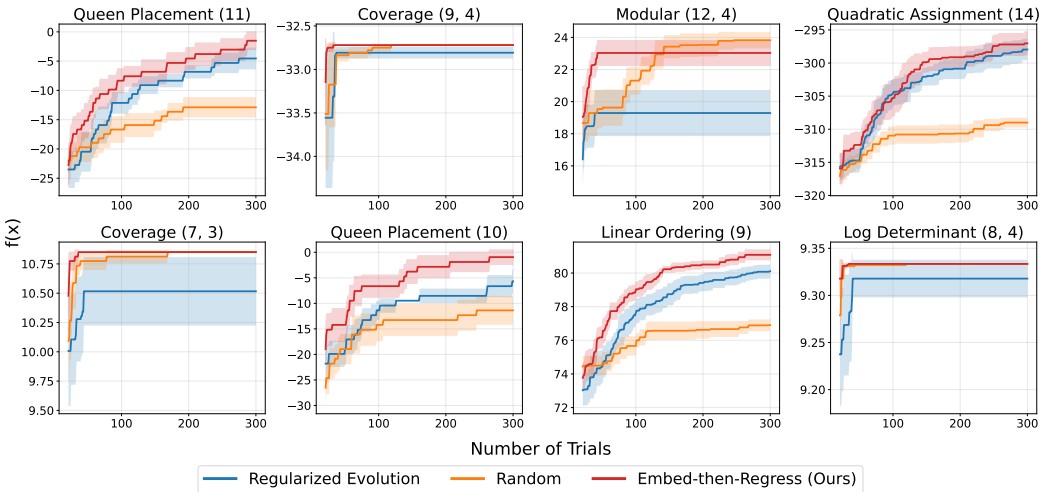

Figure 4: (↑) Higher is better. Best-so-far curves across 6 randomized combinatorial problems. Title parenthesis $(P)$ means a permutation space of size $P$ and $(N, K)$ denotes a $\binom{N}{K}$ choice space.

While there are few previous works using GPs for e.g. permutation spaces (Deshwal et al., 2022; Oh et al., 2022), they require constructing very domain-specific kernels and complex acquisition optimizers (e.g. semi-definite programming) making them difficult to reproduce. We thus use a simpler optimizer such as Regularized Evolution (Real et al., 2019) which does not need modelling assumptions other than implementing random mutations between trials and can be used broadly (Real et al., 2020). We also empirically found this was better than other evolutionary alternatives such as NSGA-II (Deb et al., 2002) or hill-climbing.

Rather than fully optimizing the acquisition, we can simply apply best-of-many sampling by using the regressor's UCB acquisition to rank sample candidates proposed by evolution, and suggest only the best. In Figure 4, we see that this boosts exploration over the original Regularized Evolution, which can often get stuck at local optima early on.

## 4.2 ABLATIONS

In this subsection, we ablate different effects on the model's prediction ability, which directly affects optimization performance.

**String Embedder Size:** In Figure 5, we see that the size of the pretrained string embedder has a monotonic influence on the predictive performance over BBOB evaluations. As we vary the T5 embedder sizes (Small, Large, XL), there is a clear trend across all predictive metrics computed over normalized $y$-values. These metrics consist of negative log-likelihood (NLL), mean average error (MAE), R-Squared, and mean absolute calibration error (MACE) (Chung et al., 2021).

It is interesting to note that larger encoders, which are pretrained over mostly English text, lead to better predictive performance over BBOB representations which do not contain any English words. Considering that the embedder's weights are also frozen, this trend potentially suggests that larger language models inherently provide better features even for numeric data formats.

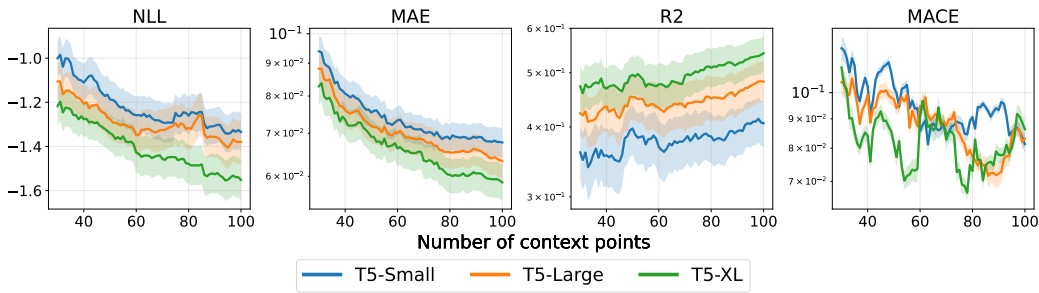

Figure 5: $(\downarrow, \downarrow, \uparrow, \downarrow)$ are better, respectively. Number of historical context points vs predictive metrics on unseen points over unseen BBOB function trajectories, while varying string embedder sizes. Solid line denotes mean over 10 test functions and error bars denote standard deviation.

**ICL Transformer Size:** In Figure 6, we find that the ICL Transformer size also plays a role, where higher layer sizes lead to better predictive outcomes. In contrast to the string embedder, here the ICL model's weights are trainable, and thus larger models can potentially possess higher capacities and better inductive biases to train over the same offline data.

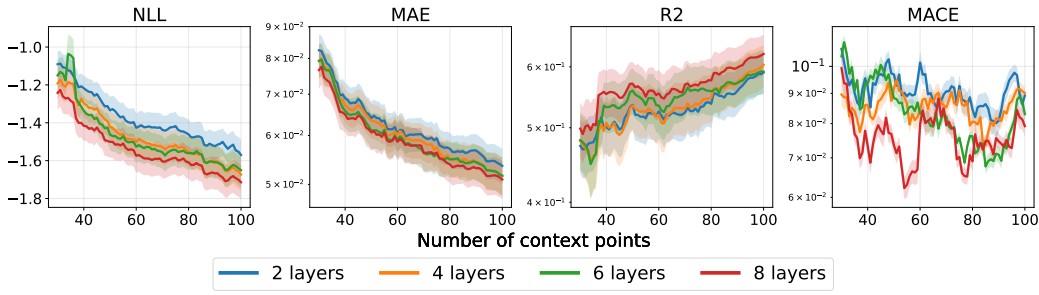

Figure 6: Analogous setting to Figure 5, while varying the number of attention layers of the ICL Transformer.

Overall in both cases for Figures 5 and 6, we verify in-context regression occurring for different test functions, where more context points leads to better predictive performance.

## 5 CONCLUSION AND FUTURE WORK

Our method, *Embed-then-Regress*, demonstrates the versatility of using string-based in-context regression for Bayesian Optimization over a variety of problems. We have shown it to obtain comparable results against industry-standard GP baselines and allow flexibility in more esoteric spaces such as permutations and combinations.

As strings are significantly more flexible representation formats of different data types, an ambitious and exciting direction is to pretrain a unified in-context regression model over multiple different domains, in order to obtain a "universal" in-context regressor. Furthermore, our method is not limited only to Transformers for in-context regression; it may be possible to additionally create a string-based GP sending string embeddings inputs to a kernel.

Further possible applications include prompt optimization (Fernando et al., 2024) and code search (Romera-Paredes et al., 2023), areas which still predominantly use zeroth-order evolutionary algorithms or even random search, which can be very sample inefficient compared to Bayesian Optimization. Additionally, outside of blackbox optimzation problems which are stateless with respect to inputs, it is worth investigating whether such methods are applicable for process-based reward modelling (Lightman et al., 2024) and tree search-based approaches (Yao et al., 2023) for stateful environments in language modelling.

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

# APPENDIX

## A    MODEL DETAILS

The full list of hyperparameters:

- ICL Transformer Size: 1024 feature dimension, feedforward projection outputs of 4096, and 8 layers of multi-headed attention with 16 heads.
- String-Embedding: We use the T5-XL encoder. Strings were clipped to a maximum of 400 tokens, using the SentencePiece tokenizer (Kudo & Richardson, 2018) with a vocabulary of 32000 subword tokens.
- Training: Effective batch size of 16, learning rate of $5 \times 10^{-4}$, weight decay of $10^{-5}$, gradient clipping of 0.5. A fixed number $T \geq 100$ total trials were always placed in the context window, with the number of history trials $t'$ sampled between $[10, T - 10]$ and the rest were target points for loss computations.
- Inference: UCB coefficient $\sqrt{\beta} = 1.8$.

For traditional optimization tasks, "Firefly" acquisition optimizer's maximum evaluation budget was $10,000$ for both our method and Vizier's, comparable to the regular default budget of $75,000$. Early results showed no difference in end-to-end optimization performance.

For combinatorial tasks, Regularized Evolution used an initial population size of 50, and a tournament size of $7 \approx \sqrt{\text{population size}}$, as prescribed in (Real et al., 2019). When it was augmented by the acquisition, we chose the highest scoring proposal out of 5 samples as the final candidate for evaluation.

## B    BENCHMARKING

For every algorithm and objective pair, we run 20 seeds and plot the best-so-far median with (25-75) percentiles as error bars.

### B.1    BBOB

Our train-test split is performed equally across all landscape types[1] (separable, low/moderate conditioning, high conditioning + unimodal, multi-modal with global structure, multi-modal with weak global structure):

- Train: {Sphere, Ellipsoidal, Rastrigin, AttractiveSector, StepEllipsoidal, Ellipsoidal, Discus, BentCigar, Weierstrass, Schwefel, Gallagher101Me}
- Test: {BuecheRastrigin, LinearSlope, RosenbrockRotated, SharpRidge, DifferentPowers, SchaffersF7, SchaffersF7IllConditioned, GriewankRosenbrock, Gallagher21Me, Katsuura, Lunacek, NegativeSphere, NegativeMinDifference, FonsecaFleming}

For transformations, we applied the following, given an initial function $f : \mathbb{R}^{\dim} \to \mathbb{R}$:

- Shifting: Uniformly samples a shift $c \in \mathbb{R}^{\dim}$, and transform $f(x)$ into $f(x - c)$.
- Rotation: Uniformly samples an orthonormal matrix $\mathbf{R} \in \mathbb{R}^{\dim \times \dim}$ and transforms $f(x)$ into $f(\mathbf{R}x)$.
- Discretization: Each parameter is randomly chosen to remain continuous, or either a `DISCRETE` or `CATEGORICAL` parameter, whose feasible points are selected over a uniform grid between the original bounds $[-5, 5]$, with the number of feasible points uniformly selected from 2 to 16.

The offline training dataset consisted of 1M tasks over sampled training objectives (along with random transformations) with each trial randomly sampled from its corresponding search space.

---

[1] https://numbbo.github.io/coco/testsuites/bbob

## B.2 COMBINATORIAL

We implemented both objective functions and evolutionary algorithms in PyGlove (Peng et al., 2020), a framework for evolutionary and combinatorial optimization.

**Permutation:** Let $x$ be a permutation of $[n] = \{1, 2, \ldots, n\}$ where $x^{(i)}$ denotes the permutation index at position $i$.

- Travelling Salesman: $f(x) = -\sum_{i=1}^{n-1} \|\text{City}(x^{(i)}) - \text{City}(x^{(i+1)})\|_2$ where each city's location is randomly sampled from $\mathbb{R}^2$
- Flowshop Scheduling: $f(x) = -\sum_{i=1}^{n} C_{i,x^{(i)}}$ where $C \in \mathbb{R}^{n \times n}$ is a random set of costs.
- Linear Ordering: $f(x)$ is the upper-triangular sum of the corresponding matrix after applying a permutation of rows and columns on $W \in \mathbb{R}^{n \times n}$ using $x$.
- Quadratic Assignment: $f(x) = -\text{Trace}(WPDP^\top)$ where $W, D \in \mathbb{R}^{n \times n}$ are random weight and distance matrices, respectively, and $P$ is the permutation matrix associated with $x$.
- N-Queens: A generalization of the classic 8-Queens problem, in which the $i$-th queen is placed on $(i, x^{(i)})$ and $f(x)$ is negative of the number of pairs of queens which diagonally attack each other.

**Choices:** Let $\text{Ind}_k$ denote the collection of all $k$-sized subsets of $[n]$. We may represent $x \in \text{Ind}_k$ as a set of $k$ indices.

- Modular Function: $f(x) = \sum_{i \in x} w^{(i)}$ where $(w^{(1)}, \ldots, w^{(n)}) \in \mathbb{R}^n$ are random weights.
- Coverage Function: Let $E_1, \ldots, E_n$ be random covers, i.e. subsets of $[n]$ and $(w^{(1)}, \ldots, w^{(n)}) \in \mathbb{R}^n$ be random weights. Let $\text{UnionCover}(x) = |\cup_{i \in x} E_i|$. Then $f(x) = \sum_{j \in \text{UnionCover}(x)} w^{(j)}$
- Log Determinant: Given a randomly sampled positive semi-definite matrix $M \in \mathbb{R}^{n \times n}$, $f(x) = \log \det(M')$ where $M' \in \mathbb{R}^{k \times k}$ is the minor of $M$ using the indices from $x$.

Offline data collection was done similarly to BBOB (i.e. random problem with random trial sampling), to generate 1M tasks and corresponding trajectories.

## C    EXAMPLE STRING REPRESENTATIONS

Below, we provide some string representations of inputs $x$ from different optimization tasks.

| Benchmark | Example $x$ | Example $m$ (if applicable) |
|---|---|---|
| Traditional (Continuous) | `x0:-0.3`
`x1:4.5`
`x2:-1.2`
`x3:"-4.1"` | `x0:DOUBLE,[-5,5]`
`x1:DOUBLE,[-5,5]`
`x2:DOUBLE,[-5,5]`
`x3:DOUBLE,[-5,5]` |
| Traditional (Categorical) | `x0:"-1"`
`x1:"-1"`
`x2:"3.5"` | `x0:CATEGORICAL,["-5",-4",...,"4","5"]`
`x1:CATEGORICAL,["-5",-3","-1",...,"3","5"]`
`x2:CATEGORICAL,["-5","-4.5",...,"4.5","5.0"]` |
| Combinatorial (Permutation) | `permutation:[0, 4, 2, 1, 3]` | `task:"Permutation"`
`size:4` |
| Combinatorial (Choice) | `choice:[1, 3]` | `task:"Choice"`
`size: 4-choose-2` |

