# OpenReview forum: "Predicting from Strings: Language Model Embeddings for Bayesian Optimization"
_ICLR.cc/2025/Conference — ICLR 2025 Conference Withdrawn Submission_

### Official Review · Reviewer_NkBm · 2024-11-03

**Soundness:** 2
**Presentation:** 2
**Contribution:** 2
**Rating:** 3
**Confidence:** 5

**Summary:**

This paper proposes a method called Embed-then-Regress that uses a pre-trained language model to embed string representations of solution candidates for a given optimization problem followed by estimating the mean and uncertainty of the solution candidates using a transformer-based Neural Process (NP) model that is trained on a set of tasks related to the problem being solved. The estimates from the NP are then used to compute UCB acquisition function values for Bayesian optimization.

**Strengths:**

1. The paper is written clearly and easy to follow.
2. The problem being addressed is important and a good solution has the potential to be impactful in several fields.

**Weaknesses:**

1. While interesting, my biggest concern with this work is that none of the ideas presented seem novel.
    (a) Using language model embeddings for latent-space Bayesian optimization is not new [1, 2]. However, no comparisons have been demonstrated with prior works.
    (b) Training an in-context regressor using a transformer was done in [3] and then later applied to BO in [4]; this work seems to swap the transformer architecture from [3, 4] (specifically the Riemann distribution mechanism) for an NP. If that is the major contribution, then a detailed comparison should be shown. Also, the authors should cite the original NP papers [5, 6].
    (c) Meta-learning for BO, again, is not novel [7, 8, 4]. Given that this method requires pre-training on several related tasks, it seems natural that the compared baselines should also be exposed to additional training for a fair comparison, but no such analyses are included.
2. Regarding the "Traditional Optimization" experiments, it is unclear given the results whether using LM embeddings provides any substantial benefit over existing approaches, especially given that stronger baselines [9] have not been compared against. I do, however, see the value in using embeddings for combinatorial problems but, as mentioned in my previous point, comparisons with relevant existing methods [1, 4] should be included to come to an informed conclusion.
3. In the conclusion and future work section, the authors also mention that _"an ambitious and exciting direction is to pretrain a unified in-context regression model over multiple different domains, in order to obtain a “universal” in-context regressor."_ Please see [10, 11].

[1] Kristiadi et al., 2024. A Sober Look at LLMs for Material Discovery: Are They Actually Good for Bayesian Optimization Over Molecules?
[2] Ranković et al., 2023. BoChemian: Large language model embeddings for Bayesian optimization of chemical reactions.
[3] Müller et al., 2022. Transformers can do bayesian inference.
[4] Müller et al., 2023. PFNs4BO: In-Context Learning for Bayesian Optimization.
[5] Garnelo et al., 2018. Conditional Neural Processes.
[6] Garnelo et al., 2018. Neural Processes.
[7] Wang et al., 2024. Pre-trained Gaussian Processes for Bayesian Optimization.
[8] Fan et al., 2023. HyperBO+: Pre-training a universal prior for Bayesian optimization with hierarchical Gaussian processes.
[9] Cowen-Rivers et al., 2021. HEBO Pushing The Limits of Sample-Efficient Hyperparameter Optimisation.
[10] Chen et al., 2022. Towards Learning Universal Hyperparameter Optimizers with Transformers.
[11] Song et al., 2024. OmniPred: Language Models as Universal Regressors.

**Questions:**

1. In all experiments, am I correct in understanding that the search space is pre-defined over a fixed set of discrete points for which the embeddings are already computed? If that is not the case and the search is, instead, over the entire embedding space, please clarify how acquisition optimization, which will result in an embedding, provide the final text of the solution, i.e. how do you go from the embedding back to the string?
2. From L219, it appears that each optimization round requires a large number of forward passes. How expensive is the presented procedure compared to the baselines in terms of wall-clock time?

---

> ### Author Response · Authors · 2024-11-26
> **Response**
>
> We thank the reviewer for the detailed feedback. Inline responses below:
>
> ## Weaknesses
> 1. Novelty
>     * (a) Comparisons to [1, 2]: While we agree that [1,2] have a similar design, these works have been very specific to chemistry and biology problems, and have also used domain-specific embedders (e.g. "Chem-T5") but have not been tested on traditional and general blackbox optimization. Our novelty comes from understanding the performance of LLM embeddings in more general scenarios.
>     * (b) Our emphasis isn't on the specific ICL regressor, as one could indeed interchange it with [3,4]. Ours indeed is a simple reimplementation of (Nguyen, 2022). We will cite [5,6].
>     * (c). "Meta-learning isn't novel" - But the term "meta-learning" is quite broad and there are numerous different ways to perform meta-learning for BBO. We don't think the existence of [7, 8, 4] means there are no new methods to be discovered. We also discussed the nuanced comparisons between different meta-learning methods in our related works section, and believe that our method has the distinct advantage of avoiding tabular featurization by representing inputs as raw strings.
>
> 2.  "LM embeddings providing benefits" - We indeed don't think (and neither do we claim so) that the benefit comes from massively improved performance over traditional algorithms, especially since the model was trained over synthetic data, and the traditional field of blackbox optimization is the most competitive domain with multiple strong "specialist" methods. We believe the real benefit to our work is moving towards a generalist form of regression over multiple different domains.
>
> 3. Comparison with [10, 11]: We specifically discussed comparisons to [10, 11] in our related works section. [10] requires a very customized tokenization that only works with tabular data, while [11] is not in-context, making it difficult to use in optimization scenarios. Our method can be seen as "trying to combine the best of both worlds", i.e. allowing free-form strings while still providing strong ICL and uncertainty quantification capabilities.

---

> > ### Comment · Reviewer_NkBm · 2024-11-27
> >
> > Thank you for your response.
> >
> > I disagree that [1] is specific to chemistry simply because of using a domain-specific embedding model. The focus should be on their method, which seems quite generic, and not on the application area. Given [1], we already learnt that LM embeddings can, indeed, be used for BO. In the present work, that seems to be a central takeaway, especially since the claim is not in outperforming stronger baselines. This is my main concern about the lack of novelty -- we are not learning anything new.
> >
> > I see now the mention of [10] and [11] in your Related Work, thank you. However, it remains difficult to agree with the claim that this method provides the "best" of both worlds, since there isn't a comparison with stronger baselines.
> >
> > To me, therefore, this work is currently not ready for publication, and I will maintain my score.

---

### Official Review · Reviewer_9mVg · 2024-11-03

**Soundness:** 3
**Presentation:** 3
**Contribution:** 3
**Rating:** 6
**Confidence:** 2

**Summary:**

The authors proposes an icl regressor for black-box optimization with four desirada in mind:
1. Pretrainable
2. Flexible representation of inputs
3. Long-range in-context regression
4. Support diverse objective scales

The main novelty in the proposed in-context regressor is using a text encoder to represent inputs $x$ as a fixed dimensional vector - by first converting to a json str $s$ and then embedding $s$ using T5 encoder. Another novel proposal is to use a text embedding model to encode metadata associated with the function being optimized as feature in in-context regression.


On both traditional black-box optimization problems and combinatorial optimization problems, the authors find their pre-trained regressor combined with ucb acquisition is comparable and sometimes better than baselines that are based on GP regressors (Vizier, GP-bandit), as well as other baselines including Regularized Evolution, random, and quasi-random search. Ablation studies show that scaling both the input embedder and the decoder leads to improvements in BBO performance.

**Strengths:**

1. Method is simple and flexible.
2. Presentation of the method is mostly clear and easy to follow.
3. Discussion of related work and motivation is clear.
4. Conducted comprehensive experiments on synthetic functions and combinatorial optimization problems.

**Weaknesses:**

1. Lacking discussion of the effect of a fixed dimensional representation of inputs x regardless of how many inputs there are. This could in principle be a bottleneck, but maybe in practice the embedding dimension is large enough that this is not an issue. Some discussion or analysis of how performance changes with the number of inputs to the function being optimized would be helpful.

2. While easy to follow, some details about the method/experiment are missing from the paper. Please see questions.

**Questions:**

1. Details of the method/experiments:
a) What meta-data is actually used in the experiments?
b) What’s the architecture of the trainable projection used to convert y into an embedding?
c) Are inputs x scaled as well or left unchanged, before converting to json?

---

> ### Author Response · Authors · 2024-11-26
> **Thank You**
>
> Thanks for your positive feedback on the simplicity of our method - Responses below:
>
> ## Weaknesses
> 1. Recently, (Tang et al, 2024) discovered that LLM embeddings can be quite strong for high-dimensional regression, even for objectives up to e.g. 100 parameters. Along with the T5-XL encoder dimension being 2048, we hypothesize that higher dimensional objective functions should also therefore be regressable in our ICL case as well.
>
> ## Questions:
> 1a) Since we trained over synthetic data, there was no need for metadata - we mentioned the possible use of metadata as an option if there are additional "graybox" clues to the objective (e.g. the name of the objective), which is similar to the setting seen by (Chen et al, 2022).
>
> 1b) $y$ was sent through a basic MLP (ReLU) layer, whose output dimension is $d$.
>
> 1c) In principle, since we're using language model inputs, $x$ does not need to be re-scaled, provided that enough pretraining has been done, similar to OmniPred (Song et al, 2023). Since in this paper we used cheap synthetic data for pretraining where each parameter in $x$ is bounded within $[-5,5]$, at inference, the model is "used to seeing" numbers within $[-5,5]$, and thus during inference time, we would need to apply an affine transformation from any new bound $[a,b]$ to $[-5,5]$.

---

> > ### Comment · Reviewer_9mVg · 2024-12-03
> >
> > Thank you for your response, I have no further questions at this point.

---

### Official Review · Reviewer_BFQF · 2024-11-12

**Soundness:** 2
**Presentation:** 3
**Contribution:** 2
**Rating:** 3
**Confidence:** 4

**Summary:**

The work proposes "Embed-then-regress", a method for using in-context regression to perform Bayesian Optimization (BO). Specifically, the paper uses a pretrained encoder network to map points from the search space (represented as strings) to a fixed-length embedding. Then, an in-context regressor is trained using the past observations as the context set to predict the distribution of the objective value for a new query point. The proposed method was experimentally evaluated on a set of synthetic functions and classic combinatorial optimization problems.

**Strengths:**

1. The paper addresses a timely problem in machine learning, namely meta-learning or large-scale pretraining for optimization tasks (e.g. black-box, or combinatorial).
2. The paper is written well, with sufficient details and context to make it easily parsable.
3. The proposed method is evaluated on both traditional optimization and combinatorial optimization tasks.

**Weaknesses:**

The main weaknesses are novelty, justification for design decisions, and thoroughness/fairness of empirical comparisons:

1. **Novelty:** The authors claim as a contribution that they demonstrate the 'versatility of using string-based in-context regression for Bayesian Optimization'. As far as I know, this has already been shown in prior works ([`R1`, `R2`]). Specifically, these works employed LLM-based optimizers and string-based representations of the search space, and has been shown to improve efficiency for (1) traditional regression, (2) Bayesian optimization, (3) combinatorial optimization (TSP). Can the authors crisply articulate the novelty of their method with respect to these works?
2. **Design decisions:** This relates the motivation for using a pretrained embedding network + an ICL regressor on top. An alternative approach could be to finetune a pretrained language model directly to do ICL with a context set of string-based representations of past observations. To me, there are two advantages of using this alternative approach: (1) ameliorates pretraining effort (especially a large ICL Transformer), (2) avoids the need for pooling techniques to aggregate over sequence embeddings (which can lose information)
3. **Empirical concerns:**
* **3a) Weak baselines:** The authors mainly compare against GP (in Fig 3) and Regularized Evolution (in Fig 4). These are fairly toy baselines, and not appropriate for a proper comparison of the model's performance. The experiments should, at the *very least*, be compared against more competitive BO (HEBO, BOHB, Deep GP/Deep Kernel Regression [`R3`, `R4`, `R5`]) and combinatorial optimizers (population-based genetic algorithms, or local search). These are not pretrained or meta-learned. Perhaps more appropriate would be to compare against techniques with similar goals of *pretraining/meta-learning* (meta-BO methods e.g. PFNs4BO [`R6`], OptFormer (which the authors already cite) although the latter is geared towards hyperparameter tuning).

If this work were the first to propose string-based representations + in-context pretraining for optimization, I would be more inclined to agree that the limited baselines are sufficient. However, given the existence of prior works [R1-R2], I feel strongly that the empirical evaluations should be more rigorous and fairer.

* **3b) Computational time:** I would like to see the wall-clock time or some metric of computational complexity. In the appendix, the authors mentioned they used 10k proposal points for each step. If my understanding is correct, that is a large number of forward passes just to acquire one point (even with parallel forward passes). This complexity would be orders of magnitude higher than GP or RS (which would take on the order of a few minutes on a CPU for the entire search).

* **3c) Performance improvements:** The performance improvements are relatively limited. In Fig 3, GP methods match embed-then-regress on around half of the problems. These results are not convincing, considering the use of large-scale training and significantly higher computational complexity.



---

[R1] Yang, C., Wang, X., Lu, Y., Liu, H., Le, Q.V., Zhou, D. and Chen, X., Large language models as optimizers. arXiv 2023. arXiv preprint arXiv:2309.03409.

[R2] Liu, T., Astorga, N., Seedat, N. and van der Schaar, M., 2024. Large language models to enhance bayesian optimization. arXiv preprint arXiv:2402.03921.

[R3] Cowen-Rivers, A.I., Lyu, W., Tutunov, R., Wang, Z., Grosnit, A., Griffiths, R.R., Maraval, A.M., Jianye, H., Wang, J., Peters, J. and Bou-Ammar, H., 2022. Hebo: Pushing the limits of sample-efficient hyper-parameter optimisation. Journal of Artificial Intelligence Research, 74, pp.1269-1349.

[R4] Falkner, S., Klein, A. and Hutter, F., 2017, December. Combining hyperband and bayesian optimization. In NIPS 2017 Bayesian Optimization Workshop (Dec 2017).

[R5] Wilson, A.G., Hu, Z., Salakhutdinov, R. and Xing, E.P., 2016, May. Deep kernel learning. In Artificial intelligence and statistics (pp. 370-378). PMLR.

[R6] Müller, S., Feurer, M., Hollmann, N. and Hutter, F., 2023, July. Pfns4bo: In-context learning for bayesian optimization. In International Conference on Machine Learning (pp. 25444-25470). PMLR.

**Questions:**

I also have some questions and minor concerns:

1. In L172-179, the authors describe the normalization process for $y$ values, but are input values (for each dimension of the search space) normalized too? This might present an issue if the search spaces between tasks exist on very different scales.
2. How exactly are pretraining tasks created/selected for the results shown in Fig 3 and Fig 4. How many pretraining tasks were included in total, and how many observations sampled from each problem?
3. The hyperparameters for regularized evolution seem inappropriate. What exactly are the hyperparameters used (for genetic operations etc), was the population size of 50 maintained throughout? That is a very small population size.
4. I understand that multiple parallel predictions can be made on a batch of query points simultaneously. But is the embedding process parallelized too?

---

> ### Author Response · Authors · 2024-11-26
> **Response**
>
> Thanks for spending the time to thoroughly read through our paper and its relation in previous work. We address specific issues below, and will take care to incorporate some of this feedback into updated drafts.
>
> ## Weaknesses
> 1.**Novelty:** R1 and R2 are completely at the mercy of service-provided LLMs and their business priorities, because they rely on emergent behaviors of LLMs. This means that performance is a function of completely uncontrollable factors, such as:
>   * Whether the service allows long-contexts.
>   * Which data mixtures were used during pretraining / post training
>   * What the service provider decided for architecture size.
>
> Since LLM providers (at least those we're aware of) aren't interested in use-cases for optimization, it's impossible for an optimization researcher to improve such model capabilities themselves. At times, we've seen e.g. ChatGPT worse than random search due to these issues. In comparison, our work provides a method specifically tailored to optimization and allows tuning to the user's offline (x,y) data, which is the most significant factor to regression performance.
>
> Also, the term "LLMs for Optimization" is an extremely broad phrase, even though there are endless different ways to use LLMs for optimization, similar to "GPs for optimization" also having many different variants. We don't believe R1 or R2 existing should mean that there's no more novelty to using LLMs in various other ways.
>
> 2.**Design Decision:** On fine-tuning via ICL by simply placing strings - this is in all honesty a fine alternative, but as we discussed in our related works section:
>   * If using raw strings, this requires a presumably large pretrained model (to understand English) with a higher memory consumption, which would easily limit the context window (as seen in R2, where at most ~60 trials could fit). This issue would be exacerbated when each $x$ may also contain additional metadata.
>   * If using customized tokenizations for better compression (as in Chen et al 2022): This is fairly restrictive to tabular formats only, and has difficulty handling e.g. combinatorial data.
>
> 3a) As mentioned in our general response, we're not advocating that our method is SOTA at all when it comes to comparisons against domain-specific optimizers, rather proposing an alternative formulation which allows the use of strings for regression during optimization.
>
> 3b) Higher computational complexity: It is indeed true, that the embedding method currently has a higher inference speed. However, there are numerous ways to reduce costs over time, (using efficient attention mechanisms throughout, using alternative acquisition optimizers, etc.). We consider our current paper a promising prototype to start from.
>
> 3c) Performance improvements - As mentioned in our general statement, our goal is not to advocate for a much better performance over GP methods (we would not have shown such experiments equalling against GP otherwise), but to understand whether embedding-based regression performs reasonably, since it is a natural design and gateway for more string-based regression methods.
>
> ## Questions
> 1. Note that if the model was pretrained on e.g. parameter bounds of [-5,5] only, it's always possible at inference time to warp any other bound down to [-5,5], with e.g. an affine scaling. Eventually however, warpings should not be needed once the model has been trained over multiple different problems with different space bounds.
> 2. We assume that the synthetic dataset is "infinite sized", since it is cheap to generate. For practical purposes, we generated an offline dataset of 1M trajectories (each with e.g. 300 random trials), where each trajectory's objective was formed by a cartesian product of (BBOB training function, shifting, categorization).
> 3. The mutations were uniformly random (e.g. for permutations, select two random indices and swap them), and the population size of 50 was indeed used throughout. We believe these hyperparameters were fairly reasonable (we also used a recommended tournament size of $\sqrt{\text{population}}$).
> 4. The embedding process parallelization should be automatically taken care of via typical computation graph compilation. At inference, historical embeddings can also be cached to reduce redundant computations.

---

### Official Review · Reviewer_v3oi · 2024-11-12

**Soundness:** 2
**Presentation:** 3
**Contribution:** 2
**Rating:** 3
**Confidence:** 4

**Summary:**

The paper proposes _embed-then-regress_ a paradigm to perform in-context regression for bayesian optimization. The premise is that in order to get around the varying sizes and dimensions of problem specifications and inputs, they use an embedding model to convert the string representation of each trial into a fixed dimension vector. They are then able to pass this to the regressor Transformer which has been trained to perform in-context learning, similar to Garg et al. "What can transformers learn in-context? A case study of simple function classes." They then utilize expplore-exploit techniques using black box optimization to show that this technique achieves competitive performance over several tasks (primarily from the [BBOB test suite](https://numbbo.github.io/coco/testsuites/bbob).

**Strengths:**

1. The paper is extremely well written. The problem definition is clear and they provide sufficient (if not too much) context to the problem and the algorithm.
2. The experiments are fairly extensive covering a suite of tasks.
3. The method seems to get around some fairly problematic issues with scaling bayesian optimization and achieves reasonable performance across a wide range of tasks.

**Weaknesses:**

1. Despite a thorough description of the problem, some of the description around the final black box optimization is still left out. I may have missed this, but I would have appreciated a clearer definition of what explore-exploit technique they were using.
2. I understand that the problem setup is over black box optimization techniques and therefore optimality gap is the right metric to measure. But this does not seem to be part of their proposal. The contribution revolves around regression to make a prediction on $\hat{y}$. I would like to see the accuracy of their method on just this task compared to strong baselines. The optimization is a downstream task that I expect will be the same regardless of the mechanism of regression.
3. The authors suggest in the motivation that the main problem they are trying to solve is the large/unbounded size of the inputs. I presume this becomes infeasible very quickly, but do simple techniques like 0-padding work at all?
4. I am not sure why the authors insist on projecting $y$ to $\mathcal{R}^d$ and turning the input into a $2d$ vector. Since it is a real number, can't it be left as is and concatenated to get a vector in $d+1$? Or they could simply try the mechanism of Garg et al., and stack the vectors as different tokens and therefore the problem becomes next token prediction with the input $\\{x_i, y_i\\}_{i=1}^{n-1}, \hat{x}$. Did the authors try this method? Does it perform worse?
5. Why use custom attention patterns to make predictions parallel? Doesn't the inference mechanism in Garg et al. with k-v caching produce the same effect of avoiding redundant computations?
6. The authors claim that the inference cost needs to be cheap since zeroth order optimizers need to be called several times. Isn't this a choice though? Higher order optimizers could be used which are far more sample efficient.
7. The baselines in the experiments seem fairly weak. Why are there two random baselines?
8. Has the regressor seen some examples from each family of the test set? How different are the parameter values between the train set and the test set? If they are within the same range, it is possibly that the model is simply interpolating between seen values. This would be easy to check by comparing with a stronger baseline - perhaps even a simple 2-layer MLP?
9. My main concern with the paper is the true finding is that transformers can learn how to solve regression using ICL on varying families of functions and the black box optimization and the embedding are merely applications of this finding. If this is the case, then this has already been shown along with theory in Li, Yingcong, et al. "Dissecting chain-of-thought: Compositionality through in-context filtering and learning."

**Questions:**

1. I found it very useful to interpret the problem as something like reducing the problem of regression to that of RAG where the generator is the regressor given in-context samples and the corpus is retrieving a compact embedding of the said samples from the problem description strings. Is this somewhat accurate?
2. In 3.1 is $\mathcal{X}$ assumed to be of finite dimension?
3. Embedding of metadata: The authors mention that "we can also embed this data..." but it is not clear what is done in the experiments.
4. The larger encoders leading to better results is indeed quite interesting. Is it possible that some of these datasets may have been part of the T-5 pre-training dataset?

---

> ### Author Response · Authors · 2024-11-26
> **Response**
>
> Thanks for the detailed feedback and the compliments around our paper's readability. Responses in-line, and we'll incorporate many of the clarifying questions and feedback into the paper.
>
> ## Weaknesses
> 1. We're using a standard UCB acquisition function, with a coefficient of 1.8. Thus, the acquisition score is (mean + 1.8 * std).
> 2. Our paper's purpose is to design regressors specifically towards optimization purposes. Note that this subtly means our model does not need to satisfy certain requirements for pure regression (in the data science sense), which is a also true for GP-based methods:
>     * $y$-normalization does not need to be strictly invertible or 1-to-1
>     * The regressor must be able to take in additional online observations at inference time.
>     * Uncertainty quantification is much more important than mean prediction.
> 3. Zero-padding techniques are used in PFNs4BO, but they're still very specific to tabular data and have size limitation issues when encountering categorical parameters with many possibilities.
> 4. This is a small detail that isn't too important to our paper's main point - we mapped $y$ to $\mathbb{R}^d$ simply for symmetry with the $x$ embedding, but could've also just concatenated to $\mathbb{R}^{d+1}$ as you mentioned.
> 5. This is another subtle but small detail - we can't use auto-regressive attention because the $y$-normalization has a clear separation between history and target points. Thus, the attention must be in the form of a Prefix-LM style pattern - i.e. all target points are fully dependent on history, but not dependent on each other.
> 6. Agreed - it is possible using soft-prompt optimization techniques to perform gradient-based acquisition maximization. In this paper, we used zeroth-order optimization since it's easier for comparisons and implementation.
> 7. Random search and quasi-random search are standard baselines. Note that Vizier GP is one of the strongest industry-standard Bayesian Optimization algorithms, which our method matches.
> 8. The regressor did _not_ see any of the test functions during training. The parameter ranges are the same for BBOB (i.e. every parameter is within $[-5,5]$) but note that any other search space can simply be warped to match $[-5,5]$.
>     * As for interpolation - we think that is fundamentally what regression (+ some inductive priors) actually is, however.
> 9. As mentioned in Question 2 - there are nuanced differences between standard regression and Bayesian Optimization, that require the paper be specific to either application.
>     * This is common within previous literature (e.g. one paper discusses Random Forests for regression, while paper adding modifications to be useful for optimization).
>
> ## Questions:
> 1. RAG is fundamentally about retrieval (i.e. looking over a corpus of already known facts), which is different from regression, which fundamentally about predicting new "numerical facts" from a query. So we think they're fundamentally different.
> 2. Yes the standard way (using feature based techniques) of representing $\mathcal{X}$ would have finite dimension. But note that $\mathcal{X}$ doesn't intrinsically have a canonical dimension (e.g. we could've used a linear projection to make the resulting feature have higher dimension). When expressed as strings, the notion of "dimension" no longer has meaning, in a sense.
> 3. One can embed additional information that might help prediction (e.g. if one has the name of the function, this would help prediction significantly) - we meant to say that our proposed model makes graybox optimization much easier.
> 4. Doubtful - T5 models are trained only over the C4 (i.e. web crawled data) which consists of mostly English text, and is unlikely to contain any substantial numeric data.

---

### Author Response · Authors · 2024-11-26
**General Response**

We thank the reviewers for their time and valuable feedback around the paper’s current framing. We will take this into consideration when editing the paper. We address the general concerns below:

## Perspective on Merits
The goal of this paper isn’t to claim a new SOTA meta-BBO method, and in fact, we don’t think it’s possible in a completely fair setting, since LLM embeddings as features are generally not as strong as hand-crafted tabular features specifically for regression.

What we want to emphasize is the urgent need for trainable _generalist regressors that can reasonably regress on anything_, as many new blackbox optimization problems have appeared over unstructured domains (Graybox, Prompting, Code Search, etc.).

This paper verifies that with a generalist string-based regressor with minimal synthetic pretraining, it’s at least possible to **achieve decent results against specialist regressors in the most well-studied and competitive domain of standard blackbox optimization.**

## Alternative Designs and Baseline Comparisons
Following above, our intention isn’t to push on any specific design of the ICL Transformer backbone, which is the same as (Nguyen, 2022). One can just as easily swap out the Gaussian head with a categorical histogram as (PFNs4BO).

Note that the baseline GP algorithm (Vizier) is very strong, as it is an industry-grade algorithm that’s been shown to outperform (SkOPT, BoTorch, Optuna, etc.) and roughly equivalent to HEBO for continuous spaces, while remaining SOTA for non-continuous spaces.

---

### Note · Authors · 2024-12-05

I have read and agree with the venue's withdrawal policy on behalf of myself and my co-authors.